# Stratigraphic and Paleoecological Significance of the Early/Middle Pleistocene Vertebrate Fauna of the Süttő 21 Site

Piroska Pazonyi [1,*] , Zoltán Szentesi [2,†], Lukács Mészáros [3,†] , János Hír [4,†] and Mihály Gasparik [2,†]

[1] ELKH-MTM-ELTE Research Group for Paleontology, Institute of Geography and Earth Sciences, Eötvös Loránd University, POB 137, H-1431 Budapest, Hungary

[2] Department of Paleontology and Geology, Hungarian Natural History Museum, POB 137, H-1431 Budapest, Hungary; szentesi.zoltan@nhmus.hu (Z.S.); gasparik.mihaly@nhmus.hu (M.G.)

[3] Department of Paleontology, Institute of Geography and Earth Sciences, Eötvös Loránd University, Pázmány Péter sétány 1/c, H-1117 Budapest, Hungary; lgy.meszaros@gmail.com

[4] Municipal Museum, Múzeum tér 5, H-3060 Pásztó, Hungary; hirjanos@gmail.com

* Correspondence: pinety@gmail.com

† These authors contributed equally to this work.

**Abstract:** The Süttő 21 site is a fissure fill of the freshwater limestone of the Gazda quarry in Süttő. The material was collected between 2017 and 2019, and the results are summarised in this article, with a special focus on the small vertebrate fauna of the site and its stratigraphic and paleoecological significance. The fissure fill can be placed around the Early/Middle Pleistocene boundary (ca. 1.1 and 0.77 Ma). The paleoecological analysis of the herpeto- and mammal fauna of the sequence indicates the proximity of a permanent water body. The lower part of the sequence is dominated by open habitat indicator taxa indicating a cool, dry climate. Towards the upper part of the sequence, the climate remained cool, but became wetter, and the vegetation gradually changed to forest-steppe/open forest. The fauna of the Süttő 21 site can be compared with the material of sites that are of a similar age, thus revealing taxonomic and paleoecological differences between different areas of the country. While a warm, dry climate and open vegetation can be reconstructed in the Villány Hills around the Early/Middle Pleistocene boundary, the Northern Hungarian areas had a cooler, wetter climate and a slightly more closed (sparse forest, forest-steppe) vegetation during this period.

**Keywords:** vertebrate fauna; paleoecology; stratigraphy; Early and Middle Pleistocene

## 1. Introduction

One of the most-important travertine quarries in Hungary, exploited until the present day, can be found south of Süttő village (Northern Hungary). This mine is one of the most-important Pleistocene vertebrate sites in Hungary and, through the site, Süttő 6, is the type locality of the Süttő Biochronological Phase (MIS 5) [1,2]. Since the middle of the 19th Century, vertebrate remains have been recovered from 20 sites in different parts of the mine (Cukor quarry, Hegyháti quarry, Diósvölgyi quarry, and Haraszti quarry), both from travertine and fissure fills in the rock. The sites described from the Süttő Travertine Complex were last summarized by Pazonyi et al. [3], and the age of the travertine and fissure faunas was correlated with the Danube terrace chronology by Ruszkiczay-Rüdiger et al. [4]. The age of the travertine is 2.0–1.8 Ma based on the fossil material recovered from it, while the age of most of the fissure fills correlate to the different phases of MIS 5 (Süttő 3, 7, 9, and 12). However, there are also younger (Süttő 16; MIS 2) and older (Süttő 17 and 19; 1.0–0.9 Ma) fissure faunas than the abovementioned age [3,4].

The site presented in this paper was discovered on the field day of the 20th Hungarian Paleontological Meeting in 2017. It is the first known fossiliferous fissure fill from the Gazda quarry (Figure 1). The site, called Süttő 21, is an unstratified loess fill deposited in

a 6 m-high fissure, from which Mihály Gasparik and his colleagues collected samples in several excavations between 2017 and 2019.

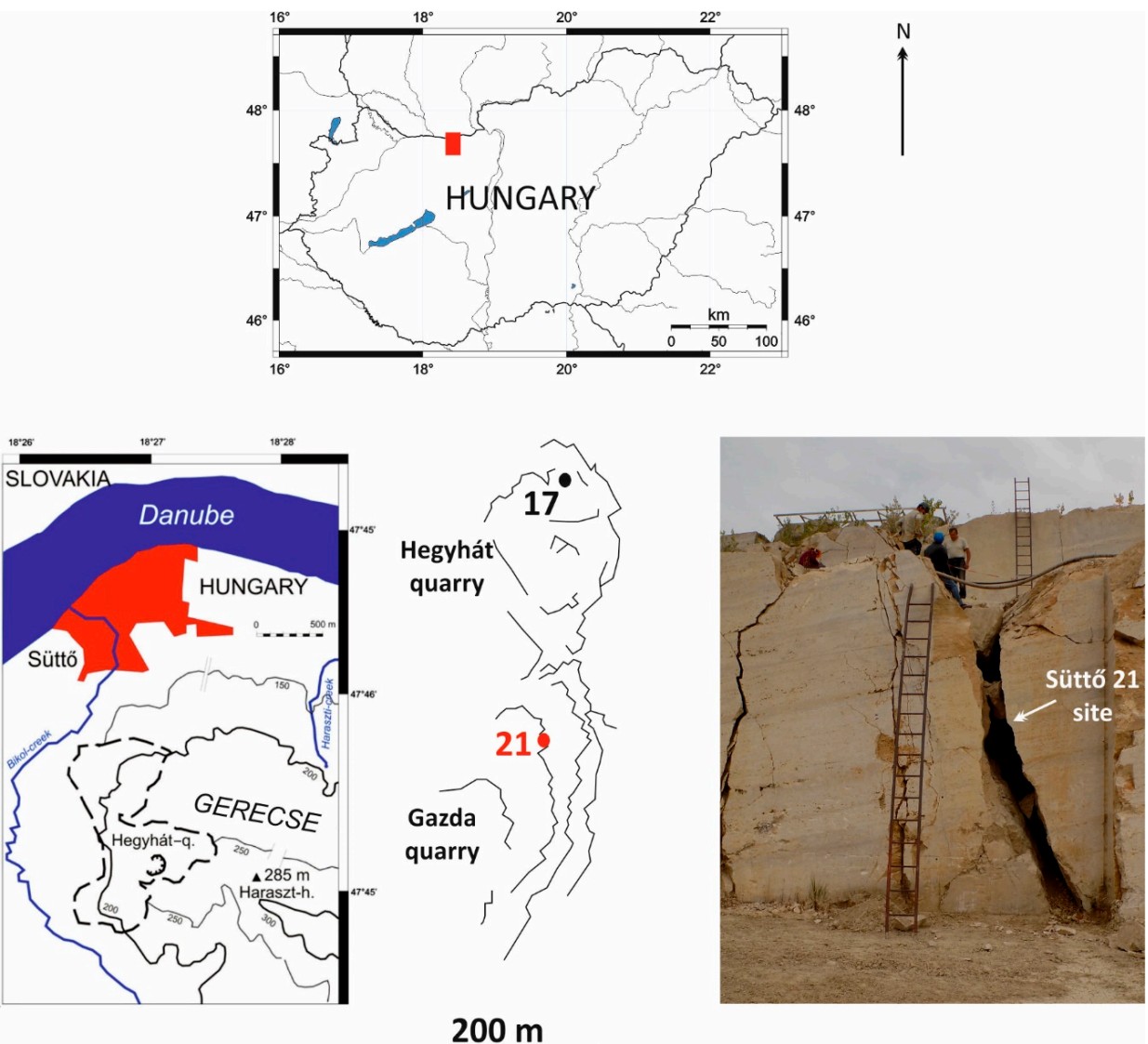

**Figure 1.** Location of the Süttő 21 site (the area marked with a red square at the top is enlarged at the bottom left). The bottom map, in the middle, shows Süttő 21 (Gazda quarry) in red and Süttő 17 (Hegyhát quarry) in black. The photo shows the 6 m-high fissure fill of Süttő 21 (modified after Pazonyi et al. [3]).

Although the analysis of the samples collected in 2019 is still ongoing, the taxonomic processing of the upper nine samples has already been performed (between 6 and 2.4 m). Fortunately, the previous collections were mainly concentrated in the lower part of the stratigraphy, so sufficient information is available for this part (between 2.4 and 0 m) also.

The main aim of this paper was to present the stratigraphic and paleoecological significance of the small vertebrate material, so the taxonomic results are only discussed for those taxa where relevant for this paper. We also aimed to compare the material of the Süttő 21 site with other sites of a similar age in Hungary, which may help to reveal climatic and vegetation differences between different areas of the country.

## 2. Material and Methods

### 2.1. Süttő 21 Site

The deposit discovered in 2017 was a 6 m-high, unstratified loess-filled fissure at the second mining level. Figure 2 clearly shows that the infilling continued below and above this level, but we were unable to collect material from these infillings because, by the year 2022, both the original and the fissures above and below the original site had been mined.

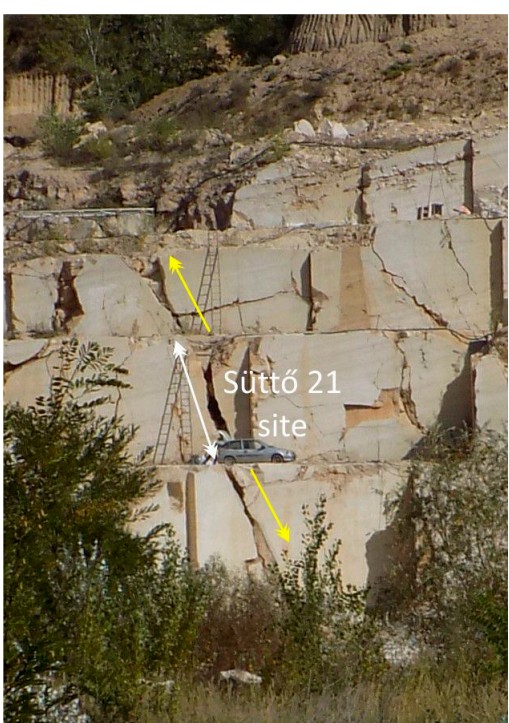

**Figure 2.** Fissure of the Süttő 21 site and its continuation on the mine levels above and below it in 2019. Due to continuous mining, the entire fissure system disappeared by 2022. The yellow arrows indicate the continuation of the Süttő 21 fissure upwards and downwards on the third and first mining levels respectively.

Between 2017 and 2019, a total of 33 samples were collected from the site during six different collections. Some of these were only test collections, where we examined the debris collected at the bottom of the fissure, but most of the samples were taken from a well-defined part of the fill. This article is based mainly on material from the 2019 excavation, when we systematically collected the entire fissure fill every 40 cm, starting from the top of the stratigraphic sequence. Since the fissure was narrower in some places and wider in others, the amount of material in each sample was different, but the smallest sample represented several bags (at least ca. 200 kg) of material, which did not cause any problems with the interpretation. Unfortunately, mainly due to the large amount of material, the material from this excavation is not yet fully processed, so the lower 2.4 m of the stratigraphic sequence is presented using material from the other excavations. Although these were smaller samples, as the material was collected from the same height in several excavations, we were able to combine them and, thus, have sufficient material to carry out the analyses (Tables 1 and 2).

**Table 1.** The herpetofauna of the Süttő 21 site stratigraphic sequence.

| Taxa | \ | \ | \ | \ | \ | \ | Sampling Sites (Meter Distances in the Studied Fissure) | | | | | | | | |
|---|---|---|---|---|---|---|---|---|---|---|---|---|---|---|---|
|  | 0.4–0.0 | 0.8–0.4 | 1.2–0.8 | 1.6–1.2 | 2.0–1.6 | 2.4–2.0 | 2.8–2.4 | 3.2–2.8 | 3.6–3.2 | 4.0–3.6 | 4.4–4.0 | 4.8–4.4 | 5.2–4.8 | 5.6–5.2 | 6.0–5.6 |
| *Triturus cristatus* | x | x |  |  |  |  |  |  |  |  | x |  | x |  | x |
| *Lissotriton vulgaris* |  |  |  |  |  |  | x | x | x | x |  | x |  | x |  |
| *Latonia gigantea* |  |  |  |  |  |  | x |  | x |  |  |  | x |  |  |
| *Pelobates fuscus* | x | x | x | x | x | x | x | x | x | x | x | x | x | x | x |
| *Bufo bufo* |  |  |  |  |  |  | x |  |  |  |  | x |  |  |  |
| *Bufotes viridis* |  |  |  |  |  | x | x | x | x | x | x | x | x | x |  |
| *Hyla arborea* |  |  |  |  |  |  | x |  |  |  |  | x |  |  |  |
| *Rana temporaria* |  |  | x | x |  |  | x |  | x | x | x | x | x |  |  |
| *Pelophylax esculentus* Group |  |  |  |  |  |  |  | x |  |  |  |  |  |  | x |
| *Lacerta viridis* |  |  |  |  |  |  | x | x |  |  |  |  | x | x | x |
| *Anguis fragilis* |  |  |  | x |  |  | x | x | x | x | x | x | x | x |  |
| *Hierophis viridiflavus* |  |  |  |  |  |  | x | x | x |  | x | x |  |  |  |
| *Hierophis gemonensis* | x |  |  |  |  |  |  |  | x | x |  |  |  |  |  |
| *Coronella austriaca* | x | x | x |  |  | x | x | x | x | x | x | x | x | x | x |
| *Elaphe* cf. *paralongissima* |  |  |  |  |  |  |  | x |  |  |  | x |  |  |  |
| *Elaphe* cf. *quatuorlineata* |  |  |  |  | x |  |  |  |  |  |  |  |  |  |  |
| *Zamenis longissimus* |  |  | x | x | x |  |  | x | x | x | x | x | x | x |  |
| *Natrix natrix* |  |  |  | x | x |  |  |  | x | x | x | x |  | x | x |
| *Natrix tesselata* |  |  |  |  | x | x | x | x | x | x | x | x | x | x |  |
| cf. *Telescopus fallax* |  |  |  |  | x |  |  |  |  |  |  |  |  |  |  |
| *Vipera* cf. *ammodytes* |  |  |  |  |  |  | x |  |  |  |  |  |  |  |  |
| *Vipera ursinii* |  |  |  |  |  |  |  |  | x |  |  | x |  |  |  |

**Table 2.** The small mammal fauna of the Süttő 21 site stratigraphic sequence with the minimum number of individuals (MNI) of each taxon.

| Taxa | \ | \ | \ | \ | \ | Sampling Sites (Meter Distances in the Studied Fissure) | | | | | | |
|---|---|---|---|---|---|---|---|---|---|---|---|---|
|  | 0.5–0.0 | 1.0–0.5 | 2.0–1.0 | 2.8–2.4 | 3.2–2.8 | 3.6–3.2 | 4.0–3.6 | 4.4–4.0 | 4.8–4.4 | 5.2–4.8 | 5.6–5.2 | 6.0–5.6 |
| Erinaceidae gen. et sp. indet. (big-sized) |  |  |  |  |  |  |  |  | 1 |  |  |  |
| *Talpa* sp. indet. |  |  |  |  |  |  |  |  |  |  | 1 | 1 |
| Desmaninae gen. et sp. indet. | 1 |  |  |  |  |  | 1 |  | 1 | 1 |  |  |
| *Beremendia fissidens* |  |  |  | 1 |  | 2 | 2 | 2 | 2 | 2 |  |  |
| *Asoriculus gibberodon* |  |  |  |  |  |  | 2 |  |  |  |  | 1 |
| *Sorex* (*Drepanosorex*) *savini* |  |  | 2 | 1 |  |  | 2 | 1 |  |  |  |  |
| *Sorex runtonensis* |  |  | 3 | 4 | 2 | 1 | 2 | 1 | 2 | 1 | 1 |  |
| *Sorex minutus* |  |  |  | 4 | 1 |  | 2 | 1 | 1 | 1 | 1 |  |
| *Sorex* sp. indet. (big-sized) |  |  |  |  |  |  |  |  |  |  |  | 1 |
| *Spalax* sp. | 1 |  | 1 |  |  | 1 | 2 | 1 | 2 | 2 | 2 |  |
| *Cricetus cricetus* ssp. | 1 | 2 | 1 | 4 | 3 | 1 | 4 | 2 | 2 | 1 |  |  |
| *Allocricetus bursae* | 1 | 1 |  | 1 |  | 1 | 2 | 5 | 1 | 2 | 1 | 1 |
| *Apodemus* cf. *sylvaticus* | 2 | 2 | 2 | 3 | 3 | 2 | 4 | 3 | 2 | 1 | 3 | 2 |
| *Sciurus* sp. |  |  |  | 1 |  |  | 4 |  |  | 1 | 2 |  |
| *Spermophilus* cf. *primigenius* | 1 | 1 | 1 | 2 | 2 | 1 | 2 | 1 | 2 | 3 | 5 | 4 |
| *Eliomys* aff. *quercinus* |  |  |  |  |  |  |  |  |  | 1 | 1 |  |
| *Muscardinus* cf. *dacicus* |  |  |  |  |  |  |  |  |  | 1 |  |  |
| *Sicista* cf. *praeloriger* |  |  |  | 4 | 2 | 2 | 2 | 2 | 2 | 3 | 6 | 1 |
| *Ochotona* sp. |  |  |  |  |  |  |  |  |  | 1 | 1 |  |
| *Pliomys episcopalis* |  |  |  |  |  |  |  |  |  | 1 | 1 |  |
| *Clethrionomys* sp. |  | 1 |  | 4 | 2 | 1 |  |  |  |  |  |  |
| *Mimomys pusillus* |  | 2 | 2 | 4 | 4 | 2 | 5 | 1 | 1 | 1 |  |  |
| *Mimomys savini* |  | 1 | 1 | 2 | 2 |  |  |  |  |  | 2 | 1 |
| *Lagurodon arankae* | 2 | 2 | 5 | 5 | 12 | 4 | 6 | 5 | 3 | 3 |  |  |
| *Prolagurus pannonicus* | 3 | 3 | 4 | 10 | 7 | 6 | 3 | 5 | 5 | 3 | 1 |  |
| *Allophaiomys praehintoni* | 2 | 3 | 3 | 7 | 2 | 2 | 3 | 7 |  |  | 1 |  |
| *Lasiopodomys hintoni* | 1 | 2 | 7 | 11 | 5 |  | 8 | 3 | 2 | 2 | 1 |  |
| *Microtus* (*M.*) *nivaloides* |  |  |  |  | 1 |  | 3 | 1 | 4 | 5 | 14 | 6 |
| *Microtus* (*M.*) *nivalinus* |  |  |  |  |  |  |  |  | 1 |  |  |  |

The collected material was washed through a 0.5 mm sieve in the Department of Palaeontology and Geology laboratory at the Hungarian Natural History Museum. The small fossils were sorted out under a stereo light microscope (Nikon SMZ 445).

The material contained seeds and terrestrial snail shells, as well as bones of fishes, amphibians, reptiles (Table 1), and mammals (Table 2). When calculating the minimum

number of individuals (MNI) values, all teeth and bones found in one sample and determined as the same species were taken into account. We examined at least how many individuals belonging to this species had to occur here for these remains to be found in the sample.

### 2.2. Sites of a Similar Age Included in the Study

Altogether, five sites were studied here from the Pannonian Basin that contained vertebrate fauna of a similar age: two were situated in the Villány Hills (Southern Hungary), while three in Northern Hungary (Figure 3).

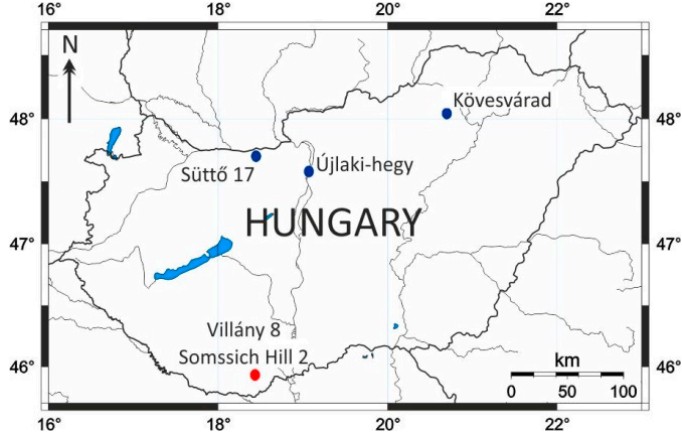

**Figure 3.** Geographical position of the Hungarian vertebrate sites of nearly the same age as the Süttő 21 site. The localities in Northern Hungary are marked in blue, while those in the Villány Hills are in red.

#### 2.2.1. Villány Hills (Southern Hungary)

Somssich Hill 2 is an 8 m-deep karst cavity with a diameter of 5 m on the present surface. It is situated within Upper Jurassic (Oxfordian) limestone at the top of Somssich Hill, to the west of Villány. The infilling sediment is a breccia with a brownish-red silty clay matrix below 4 m, whereas it becomes light yellowish-brown carbonate-cemented silt in the upper part [5,6]. The age of the locality can be correlated with the Latest Early Pleistocene (c. 1.0 Ma), the equivalent of the *Mimomys savini–Mimomys pusillus* Assemblage Zone of Kordos [7] based on the co-occurrence of the "advanced" *Allophaiomys pliocaenicus* and *Microtus* (*M.*) *nivaloides* together with *Mimomys savini* and *Mimomys pusillus* (see also [8]).

Villány 8 is a karst cavity connected to a fissure system. It is located on the southern wall of the same quarry at Templom Hill as Villány 5. The lower layers of the infilling sediment are reddish-brown clays with calcareous interbeds, whereas the upper part becomes yellowish-brown silt, similar to the Somssich Hill 2 site [2,9]. This site is the stratotype of the Templomhegyian Phase within the Biharian Stage of the local biochronological system [2,10]. The beginning of the Templomhegyian Phase can be dated to around the Early/Middle Pleistocene boundary (780 KA). Based on the absence of *Allophaiomys* and *Mimomys pusillus*, the vole fauna can be assigned to the *Mimomys savini* Biozone [7,11].

#### 2.2.2. Northern Hungary

Süttő 17 is a fissure infill with two layers of the freshwater limestone of the Hegyhát quarry in Süttő, approximately 400 m from the Süttő 21 site. The vole fauna is characterized by *Mimomys savini*, *Allophaiomys praehintoni*, *Lasiopodomys hintoni*, and *Microtus* (*M.*) *nivaloides*, while *Mimomys pusillus* is missing. Based on these dates, the age of this site is equal to the *Mimomys savini* Biozone [3].

Újlaki Hill is located to the north of Budapest as a part of the Buda Mountains. The fauna was recovered from a small karst cave filled with terra rossa, a few meters from the top of the hill, surrounded by Upper Eocene limestone [12]. *Mimomys pusillus*,

*Allophaiomys praehintoni*, *Terricola arvalidens*, and "true" *Microtus* species were found together at the site, which means that the material belongs to the *Mimomys savini–Mimomys pusillus* Assemblage Zone of Kordos [7], and its age can be correlated with the Latest Early Pleistocene (c. 1.2–1.0 Ma) [8].

Kövesvárad locality is a 5 m-high and 3 m-wide karst cavity situated to the east of Répáshuta in the Bükk Mountains. The vole material was recovered from its orange and reddish brown clay infilling [2,13]. Based on the co-occurrence of *Mimomys savini* and *Allophaiomys praehintoni* together with *Terricola arvalidens* and *Microtus* (*M.*) *nivaloides*, this site is probably contemporaneous with Újlaki Hill and Somssich Hill 2.

## 3. Biostratigraphic Results

The vole fauna of the Süttő 21 site is quite rich, with a total of 10 taxa, of which *Mimomys pusillus* and *Mimomys savini* are the zonal index species. In addition to these, two steppe lemming species, *Lagurodon arankae* and *Prolagurus pannonicus*, are very common, as are species belonging to the *Allophaiomys–Microtus* evolutionary lineage (*Allophaiomys praehintoni*, *Lasiopodomys hintoni*, *Microtus* (*M.*) *nivaloides*). Less common taxa are *Pliomys* sp., *Clethrionomys* sp., and the single example of *Microtus* (*M.*) *nivalinus*.

Based on its vole fauna, the Süttő 21 site can be dated to the so-called Mid-Pleistocene climatic revolution. During this period (1.2–0.8 Ma), the Earth's orbital ice-age cycles intensified, lengthening from ~40 ky to ~100 ky, and became markedly asymmetric. After this transition, the climate became drier and cooler, increasing the areas covered with open vegetation. The fauna of Süttő 21 represents the last phase of the transitional period, characterized mainly by the appearance of *Microtus* (*M.*) *nivalinus*, but also *Lasiopodomys hintoni* and *Prolagurus pannonicus*. Such faunas have been associated with the end of the Matuyama paleomagnetic chron (e.g., Karai-Dubina, Petropavlovka 2) [14]. Overall, the vole fauna is very similar to the late Early Pleistocene (0.9 Ma) Somssich Hill 2 material [6], but the Süttő 21 fauna lacks *Terricola arvalidens*, which dominates the Somssich Hill 2 material, and *Allophaiomys pliocaenicus*.

The situation is similar for shrews. In most parts of the sequence (except the uppermost strata), the faunal composition is very similar to that of the Soricidae assemblage of the Somssich Hill 2 site. The dominant species are *Beremendia fissidens* and *Sorex runtonensis*, with abundant *Sorex* (*Drepanosorex*) *savini* and *Sorex minutus*, while *Asoriculus gibberodon* is also present with a low number. *Neomys newtoni* is absent, but this is not surprising, as the species was not widespread yet in Central Europe at the end of the Early Pleistocene. Its sparse occurrence at Somsich Hill is the first report of this species from Hungary.

However, there is a conspicuous absence of *Crocidura* species, one of the most-abundant genera at the Somssich Hill 2 site. This would suggest that the age of the assemblage may be older than when the *Crocidura* shrews migrated from Africa to Europe, but the other Pleistocene faunal elements preclude the site pre-dating the "*Crocidura* date" (MN16/17 zonal boundary). Thus, the absence of *Crocidura*, like the variation in the vole fauna, is probably due to ecological reasons.

The mammalian fauna of the Süttő 21 site is very similar to that of the late Early Pleistocene Untermassfeld fauna [15,16] also, although the latter, probably due to paleobiogeographical reasons, lacks the lagurins (*Lagurodon arankae*, *Prolagurus pannonicus*) typical of Süttő 21. Since, in contrast to Süttő 21, magnetostratigraphic analysis of the Untermassfeld fauna was possible, we know that a paleomagnetic reversal occurred at this site, which is associated with the Jaramillo subchron (1.05 Ma), mainly based on the evolutionary level of *Microtus* (*Allophaiomys*) *thenii* [16]. The Süttő 21 site is certainly younger than Untermassfeld, since in addition to *Lasiopodomys hintoni* and *Allophaiomys praehintoni*, which have very similar morphologies to *Microtus* (*Allophaiomys*) *thenii*, two "true" *Microtus* species (*Microtus* (*M.*) *nivaloides*, *M.* (*M.*) *nivalinus*) also occur at this site (Figure 4). Further studies will be needed to clarify whether the morphology of *Lasiopodomys hintoni* and *Allophaiomys praehintoni* is a continuous transition and how these species relate to each other and *Microtus* (*Allophaiomys*) *thenii*.

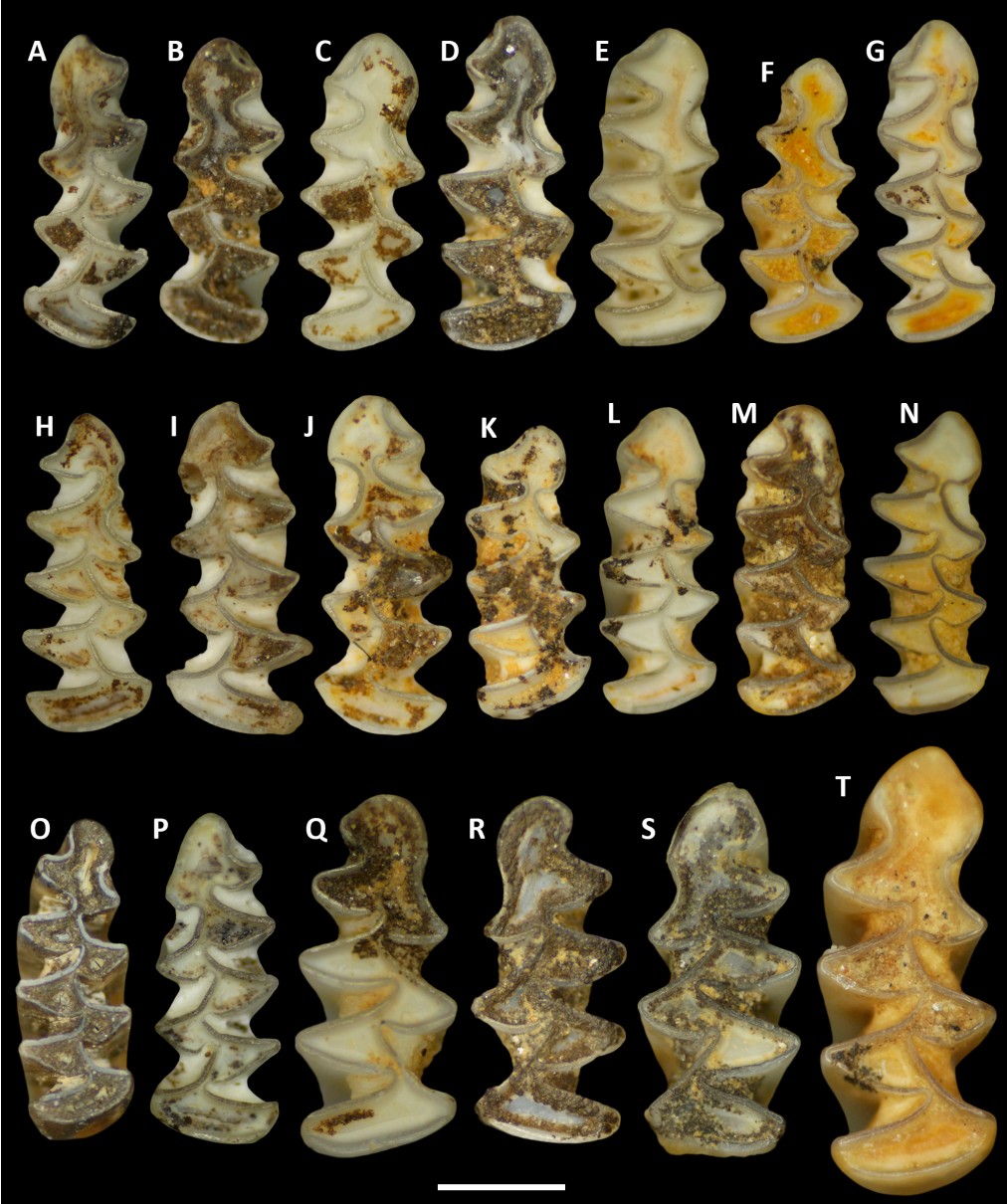

**Figure 4.** Lower first molars (m1) of some typical vole species of the Süttő 21 site. (**A–G**) *Allophaiomys praehintoni*; (**H–N**) *Lasiopodomys hintoni*; (**O**) *Microtus* (*M.*) *nivaloides*; (**P**) *Microtus* (*M.*) *nivalinus*; (**Q–S**) *Mimomys savini* (with root); (**T**) *Mimomys savini* (rootless form). The scale bar is 1 mm.

It should be noted here that the following genus or subgenus of "*Microtus*" voles were found at the Süttő 21 site: *Allophaiomys*, *Lasiopodomys*, and *Microtus* (*Microtus*). We used the taxonomic nomenclature within the Arvicolini sensu stricto tribe based on the proposed, phylogenetic-based system of Abramson et al. [17].

The fauna of small mammals in the upper part of the sequence (above 4.4 m) differs from the lower part. *Microtus* (*M.*) *nivaloides* becomes dominant in the vole fauna, and *Mimomys pusillus* and *Prolagurus pannonicus*, which were previously stable, become scarce and disappear at the top of the sequence. *Lagurodon arankae* and *Allophaiomys praehintoni* also disappear, but in addition to *Microtus* (*M.*) *nivaloides*, *Microtus* (*M.*) *nivalinus* appears (Figure 5).

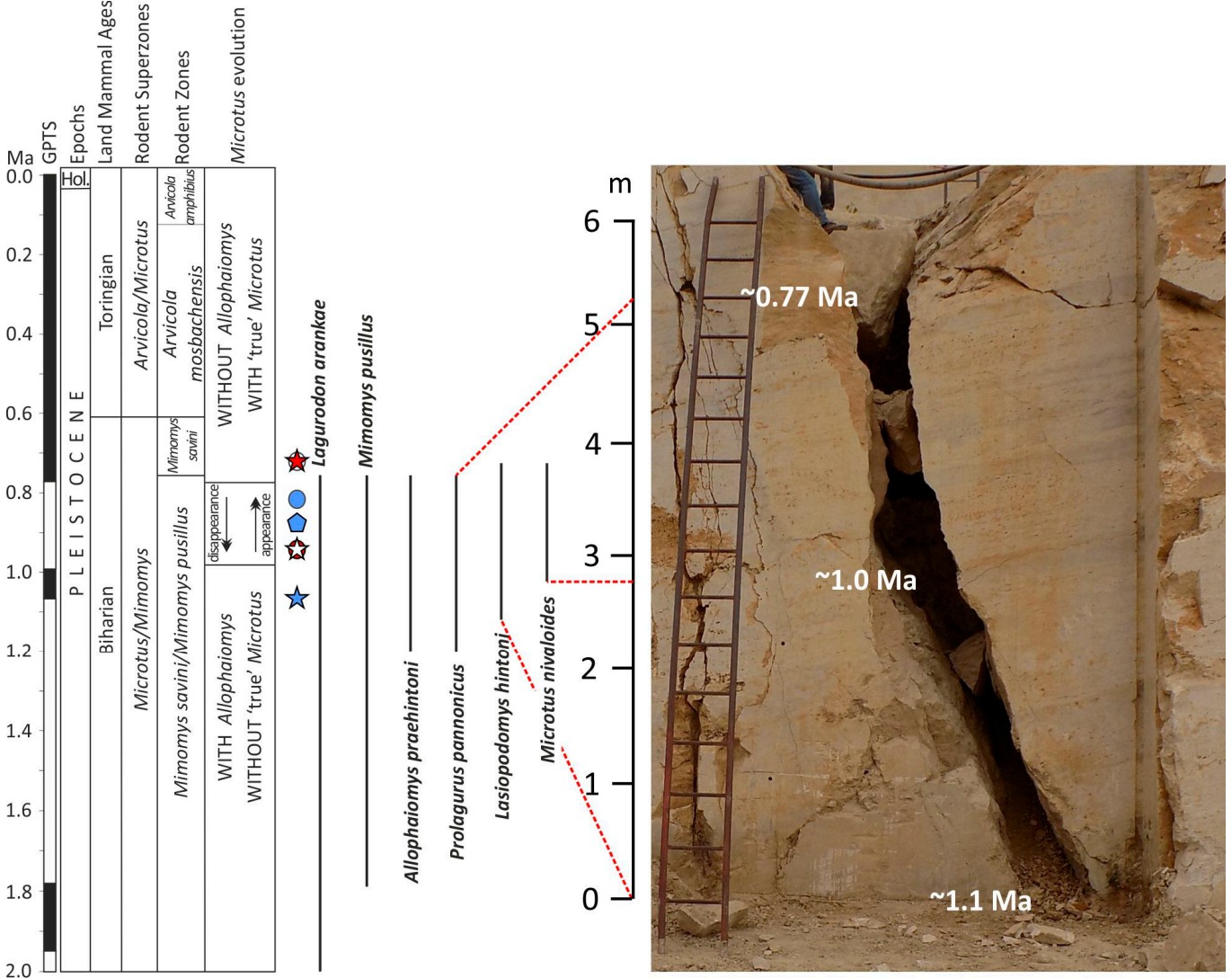

**Figure 5.** Biostratigraphic position of the 6 m-high fissure fill of the Süttő 21 site and the other Hungarian sites used for taxonomic and paleoecological comparison (modified after [9]). Legend: blue star—Újlaki Hill; white star in a red circle—Somssich Hill 2; blue pentagon—Süttő 17; blue circle—Kövesvárad; red star in a white circle—Villány 8. The exact heights of the appearance and disappearance of the age-marker vole species are indicated by a dashed red line in the section (modified after [9]).

*Microtus* (*M.*) *nivalinus* is a junior synonym of *M. ratticepoides* [18] and is one of the species typical of the last phase of the Mid-Pleistocene transition [14]. The species has been described previously from Hungary, from the Early/Middle Pleistocene Villány 8 site [9]. As this species is absent from the material of late Early Pleistocene sites (e.g., Somssich Hill 2), it can be assumed that *M.* (*M.*) *nivalinus* appeared in the area somewhat later than *M. nivaloides*.

The cap morphology of *M.* (*M.*) *nivalinus* is somewhat similar to that of the recent *Alexandromys oeconomus*. The neck is open; BRA4 is very poorly developed, but BSA4 is clearly distinguishable. The cap is slightly flattened (similar to recent *Chionomys nivalis*), but not rounded as in *M.* (*M.*) *nivaloides* and recent *A. oeconomus* (Figure 4).

In *Mimomys savini*, in the lower layers, only relatively small ($L_{mean}$ = 2.98 mm, $n$ = 3) forms with roots appear, with a high SDQ ($SDQ_{mean}$ = 165) and a low A/L ratio ($A/L_{mean}$ = 40.56, $A_{mean}$ = 1.21 mm). In contrast, the top of the sequence is characterized by rootless, large forms ($L_{mean}$ = 3.52, $n$ = 3), with a lower SDQ ($SDQ_{mean}$ = 144.45), but a higher A/L ratio ($A/L_{mean}$ = 44, $A_{mean}$ = 1.54). Rootless forms are young *Mimomys savini*, not early *Arvicola* species (Figure 4).

*Prolagurus pannonicus* is dominated by rounded anterioconid cap forms with a very simple morphology at the bottom of the sequence. At the top of the sequence (from 4 m upwards), however, the *"posterius"* morphotype becomes dominant, although the *"pannonicus"* morphotype is also present. A single specimen of the *"transiens"* morphotype was recovered from the material (between 3.6 and 3.2 m). Based on these observations, the age of the sequence is estimated to be between 1.1 and 0.8 Ma (Figure 6).

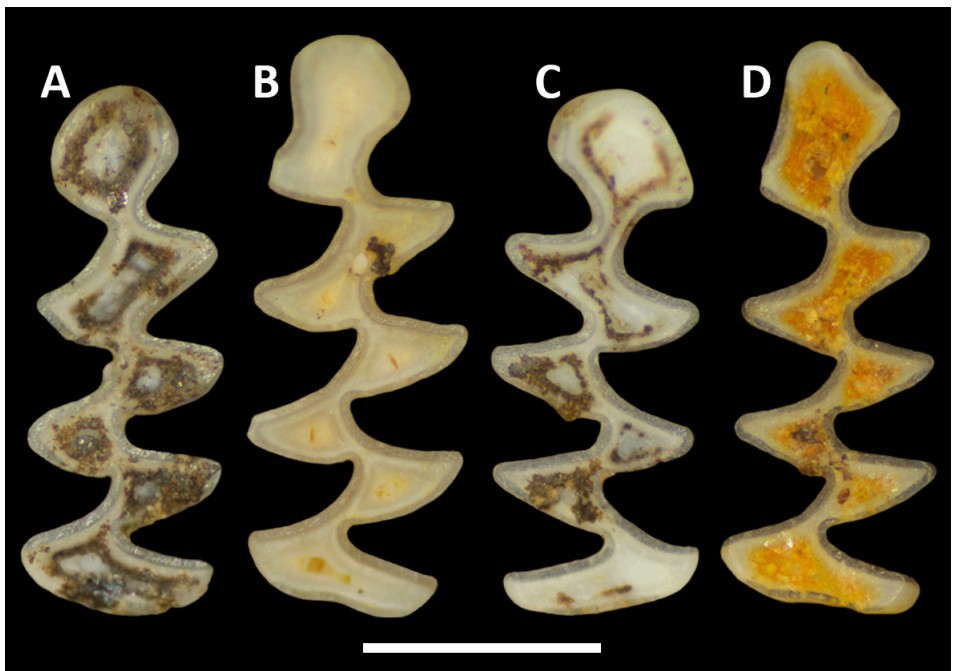

**Figure 6.** Morphotypes of the *Prolagurus pannonicus* lower first molars (m1) from the Süttő 21 site. (**A**) *pannonicus*; (**B**,**C**) *posterius*; (**D**) *transiens*. The scale bar is 1 mm.

There is also a difference between the upper and lower part of the site in the shrew fauna. At the top of the sequence, *Beremendia*, which dominate the lower levels, becomes rare and a large *Sorex* species (present only in these layers) appears.

In contrast, hamsters appear throughout the sequence of the Süttő 21 site with the same two species (*Cricetus cricetus* ssp., *Allocricetus bursae*). *Cricetus cricetus* ssp., as shown in Figure 7, is significantly smaller than *Cricetus praeglacialis* in the Middle Pleistocene material of Villány 8 and the recent *Cricetus cricetus*; however, the dimensions are definitely larger than *Cricetus nanus*. The characteristic data of this latter species are given in Table 3. A similarly small jaw with teeth m1 and m2 was also recovered from Kövesvárad (m1: L = 2.9 mm W = 1.75 mm; m2: L = 2.35 mm W = 1.95 mm), together with some large teeth, probably *Cricetus runtonensis* or *C. major*. Unfortunately, such a small number of remains are known from both Süttő 21 and Kövesvárad that we cannot give a more precise taxonomic definition for this small form.

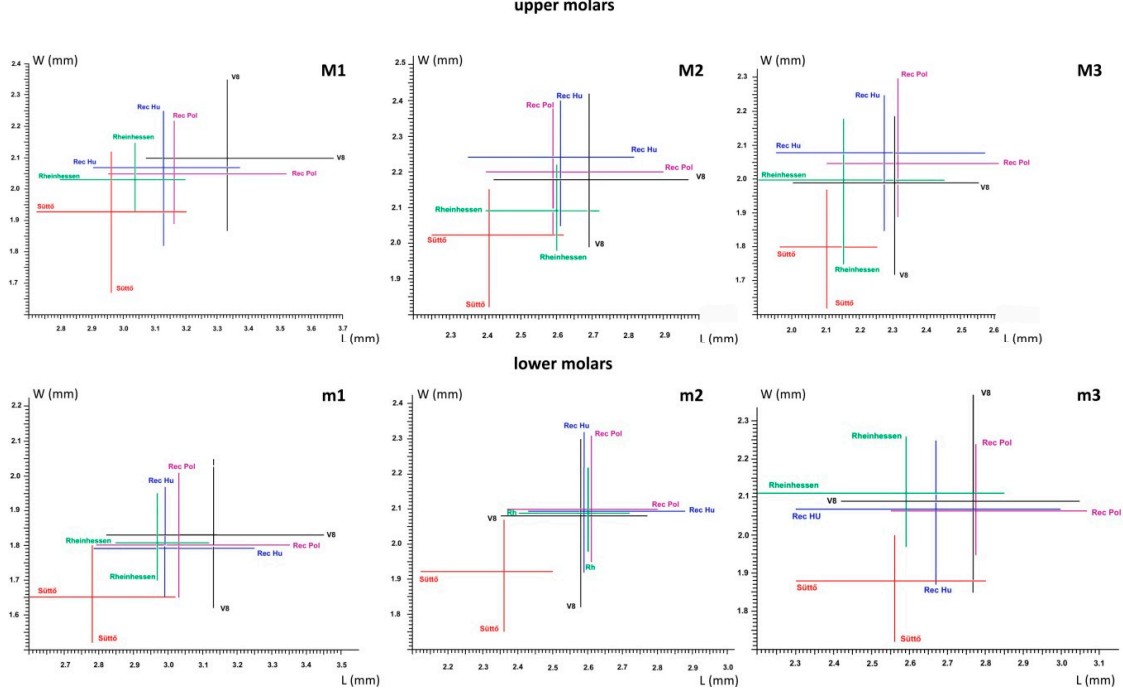

**Figure 7.** Size comparison of small *Cricetus cricetus* ssp. from Süttő 21 with other species. The sizes of each molar are shown in a separate scatter plot. V8—*Cricetus praeglacialis*, Villány 8 [19], Rec Pol—*Cricetus cricetus* recent, Poland [20], Rec Hu—*Cricetus cricetus* recent, Hungary [19], Rh/Rheinhessen—*Cricetus cricetus* recent, Germany [21].

**Table 3.** Molar dimensions of the hamster species *Cricetus nanus* from four Early Pleistocene sites in Hungary [22,23].

| *Cricetus nanus* Sites | Tooth Position | L (mm) | | | | W (mm) | | | |
|---|---|---|---|---|---|---|---|---|---|
| | | *n* | min | X | max | *n* | min | X | max |
| | M1 | 6 | 2.75 | 2.87 | 2.97 | 6 | 1.75 | 1.84 | 1.92 |
| | M2 | 4 | 1.93 | 2.21 | 2.37 | 4 | 1.64 | 1.81 | 1.95 |
| Somssich-hegy 2 | M3 | 2 | 1.74 | 1.83 | 1.92 | 2 | 1.44 | 1.46 | 1.55 |
| | m1 | 12 | 2.54 | 2.70 | 2.85 | 12 | 1.45 | 1.58 | 1.70 |
| | m2 | 7 | 2.17 | 2.25 | 2.37 | 7 | 1.72 | 1.81 | 1.92 |
| | m3 | 7 | 2.00 | 2.26 | 2.60 | 7 | 1.47 | 1.70 | 1.90 |
| | M1 | 12 | 2.85 | 2.81 | 3.0 | 12 | 1.61 | 1.74 | 1.89 |
| | M2 | 12 | 2.07 | 2.22 | 2.37 | 12 | 1.71 | 1.81 | 1.96 |
| Betfia 2 | M3 | 7 | 1.75 | 1.87 | 1.96 | 7 | 1.55 | 1.62 | 1.68 |
| | m1 | 31 | 2.41 | 2.60 | 2.85 | 31 | 1.39 | 1.52 | 1.68 |
| | m2 | 46 | 1.97 | 2.12 | 2.27 | 46 | 1.54 | 1.71 | 1.88 |
| | m3 | 34 | 2.01 | 2.23 | 2.46 | 34 | 1.48 | 1.71 | 1.95 |
| | M1 | 32 | 2.50 | 2.74 | 3.05 | 32 | 1.58 | 1.72 | 1.87 |
| | M2 | 30 | 1.88 | 2.05 | 2.18 | 30 | 1.65 | 1.75 | 1.83 |
| Osztramos 14 | M3 | 22 | 1.71 | 1.84 | 1.96 | 22 | 1.40 | 1.58 | 1.68 |
| | m1 | 11 | 2.55 | 2.67 | 2.85 | 11 | 1.44 | 1.55 | 1.64 |
| | m2 | 13 | 2.00 | 2.11 | 2.20 | 13 | 1.61 | 1.69 | 1.82 |
| | m3 | 15 | 2.04 | 2.22 | 2.56 | 15 | 1.53 | 1.64 | 1.77 |
| | M1 | 11 | 2.30 | 2.41 | 2.52 | 11 | 1.30 | 1.42 | 1.47 |
| | M2 | 8 | 2.00 | 2.12 | 2.31 | 8 | 1.62 | 1.73 | 1.82 |
| Osztramos 8 | M3 | 3 | 1.64 | 1.74 | 1.81 | 3 | 1.39 | 1.41 | 1.44 |
| | m1 | 18 | 2.30 | 2.41 | 2.52 | 18 | 1.30 | 1.42 | 1.47 |
| | m2 | 17 | 1.88 | 2.09 | 2.17 | 17 | 1.47 | 1.60 | 1.72 |
| | m3 | 13 | 1.82 | 2.05 | 2.24 | 13 | 1.44 | 1.55 | 1.64 |

Based on the observed faunistic and taxonomic changes, the sequence of the Süttő 21 can be placed around the *Mimomys pusillus–Mimomys savini* and *Mimomys savini* zones, i.e., the Early/Middle Pleistocene boundary (between ca. 1.1 and 0.77 Ma) (Figure 5). The lower part of the sequence is Early Pleistocene, while the upper part probably dates to the very beginning of the Middle Pleistocene. This may be indicated by the occurrence of *Microtus* (*M.*) *nivalinus*, as this species has so far only been described from the beginning of the Middle Pleistocene in Hungary (Villány 6 and Villány 8 sites) [9]. The stratigraphic significance of the site is that we can study the changes at the Early/Middle Pleistocene boundary in a continuous stratigraphic sequence.

Large quantities of large mammal remains were found, but unfortunately, most of them are fragmentary. The intact specimens are dominated by isolated teeth, phalanges, carpal, tarsal bones, etc., which in most cases allow more or less questionable species identification only. Thus far, the following large mammal faunal elements have been identified: *Mammuthus meridionalis*, *Equus* sp. (very probably *E. altidens* or *E. suessenbornensis*), *Capreolus* sp., *Eucladoceros* cf. *giulii*, *Bison* cf. *schoetensacki*, and *Homotherium* ex gr. *H. latidens*. The identification of *Mammuthus meridionalis* and *Homotherium* is quite certain, based on the dimensions and morphological characters of some of their remains. In the case of *M. meridionalis*, the enamel thickness measured on a tooth plate fragment is 3.8–4.2 mm, and the crown height is 75 mm. The identification of the *Homotherium* is based on an upper P4. It has no trace of preparastyle, and its protocone is extremely reduced. The maximum length of the crown is 40.25 mm; the height of the paracone is 24.75 mm; the width behind the paracone is 11.67 mm. The taxonomic classification of the other species was partly based on the dimensions and overall shape of some isolated teeth and limb bone fragments, but the most-important factor was the comparison with other European sites of a similar age (e.g., La Vallonet, Redicicoli, Atapuerca-Gran Dolina, and first of all, Untermassfeld) and the species that were identified from them [24–26].

## 4. Paleoecological Evaluation

For the paleoecological evaluation of the small vertebrate fauna of the stratigraphic sequence, in addition to taxonomic processing, it was also important to determine the MNI of the species in the case of small mammals (Table 2). For the herpetofauna, no such minimum number of individuals was determined; in Table 1, only the occurrence of each taxon in the different samples is indicated.

The Süttő 21 site contains a relatively rich herpetofauna including newts (*Lissotriton vulgaris* and *Triturus cristatus*), frogs (*Latonia gigantea*, *Pelobates fuscus*, *Bufo bufo*, *Bufotes viridis*, *Hyla arborea*, *Rana temporaria*, and *Pelophylax esculentus* Group), lizards (*Lacerta viridis* and *Anguis fragilis*), colubrid snakes (*Hierophis viridiflavus*, *H. gemonensis*, *Coronella austriaca*, *Elaphe* cf. *paralongissima*, *E.* cf. *quatuorlineata*, *Zamenis longissimus*, *Natrix natrix*, *N. tesselata*, and cf. *Telescopus fallax*), and vipers (*Vipera ammodytes* and *V. ursinii*).

The remains of the spadefoot toads (*Pelobates fuscus*) are present in all samples (Table 1). This is probably due to the fact that the soil was favourable, soft, and plastic [27], throughout for these burrowing frogs. The burrowing toads such as the *Bufotes viridis* are also relatively frequent, and some remains of *Bufo bufo* also occur in two samples. Other open land-loving animals [28,29] such as green lizards and whip snakes are significantly rarer, while the uncertain remains of the cat snake (cf. *Telescopus fallax*) appear only in the 1.6–2.0 m, while the *Vipera ursinii* is known from two samples in the profile. Except for a sample from 0.8–1.2 m, the moisture-loving herpeto elements (newts, the remains of the *Pelophylax esculentus* group, and the grass and dice snakes) are everywhere. On the other hand, fish remains (otoliths) were present from this screen-washed material, that is the permanent water existed, suggesting some fragmentary *Natrix* sp. vertebrae, as well. *Hyla arborea*, *Anguis fragilis*, *Elaphe quatuorlineata*, *Zamenis longissimus*, and *Vipera ammodytes* suggest that the paleoenvironment was woody or at least woody and bushy [28,30,31]. The presence of opportunist *Rana temporaria* and the more frequent *Coronella austriaca* suggest

the presence of a mosaic, forest edge paleoenvironment. Highly adapted to various habitats, *Latonia gigantea* cannot be used as a paleoenvironmental indicator [32–36].

In summary, the composition of this paleo-herpetofauna suggests the paleoenvironment of the site Süttő 21 was an ephemeral water body (lake) and around this loose soil and woody land, which could come into contact with a steppe.

In the case of small mammals, the ecological requirements of the species were based on recent analogies [37,38], our previous research [6,39], and the results of similar paleoecological studies [40]. We classified taxa into four categories: open habitat indicator taxa, forest–shrub indicator taxa, mesophilous taxa, and taxa with unknown ecological preferences or opportunistic species (Table 4). In addition to analysing the total small mammal fauna, we also analysed the so-called steppe species, which represent about half of the small mammal fauna and are indicative of open environments and within which significant changes in the sequence can be observed due to environmental differences.

**Table 4.** Grouping of small mammal taxa according to their habitat preferences [6,37–40].

| Open Habitat Indicator Taxa | Forest-Shrub Indicator Taxa |
|---|---|
| *Sorex runtonensis* | *Talpa* sp. indet. |
| *Spalax* sp. | *Asoriculus gibberodon* |
| *Cricetus cricetus* ssp. | *Sorex minutus* |
| *Allocricetus bursae* | *Sorex* sp. indet. (big sized) |
| *Spermophilus* cf. *primigenius* | *Apodemus* cf. *sylvaticus* |
| *Sicista* cf. *praeloriger* | *Sciurus* sp. |
| *Ochotona* sp. | *Eliomys* aff. *quercinus* |
| *Lagurodon arankae* | *Muscardinus* cf. *dacicus* |
| *Prolagurus pannonicus* | *Clethrionomys* sp. |
| | *Allophaiomys praehintoni* |
| | *Lasiopodomys hintoni* |
| | *Microtus* (*M.*) *nivaloides* |
| | *Pliomys episcopalis* |
| **mesophilous taxa** | **taxa with unknown ecological preference** |
| Desmaninae gen et sp. indet. | *Beremendia fissidens* |
| *Sorex* (*Drepanosorex*) *savini* | *Mimomys pusillus* |
| *Mimomys savini* | |

As shown in Figure 8, the ecological requirements are unknown of only 16.4% of the species that make up the fauna on average, so the analysis is quite informative. In the small mammal fauna of the stratigraphy, the taxa (*Sorex* (*Drepanosorex*) *savini*, *Mimomys savini*, Desmaninae gen. et sp. indet.) that are mesophilous are present throughout in subordinate abundance, which may indicate that there was a permanent watercourse or lake in the wider environment of the site. Although the open habitat preferences taxa (*Sorex runtonensis*, *Spalax* sp., *Cricetus cricetus* ssp., *Allocricetus bursae*, *Spermophilus* cf. *primigenius*, *Sicista* cf. *praeloriger*, *Ochotona* sp., *Lagurodon arankae*, and *Prolagurus pannonicus*) dominate the small mammal fauna, the forest–shrub indicator taxa (*Talpa* sp. indet., *Sorex* sp. indet., *Asoriculus gibberodon*, *Sorex minutus*, *Apodemus* cf. *sylvaticus*, *Sciurus* sp., *Eliomys* aff. *quercinus*, *Muscardinus* cf. *dacicus*, *Clethrionomys* sp., *Allophaiomys praehintoni*, *Lasiopodomys hintoni*, and *Microtus* (*M.*) *nivaloides*) are also present throughout the sequence. The relative proportions of the two categories allowed us to distinguish between periods of more forested and more open vegetation.

To determine the extent to which the vegetation in the area was forested, we looked at the ratio of the MNI of open and forest–shrub habitat indicator taxa in each sample. To do this, we used the formula $\mathrm{MNI_{forested}}/\mathrm{MNI_{open}} \times 100$, where $\mathrm{MNI_{forested}}$ is the sum of the MNI of the species in the sample that prefer forested and shrubby environments and $\mathrm{MNI_{open}}$ is the sum of the MNI of the species that prefer open land vegetation. The vegetation closure index is 100 if $\mathrm{MNI_{forested}} = \mathrm{MNI_{open}}$, less than 100 if the vegetation was more open, and greater than 100 if the vegetation was more forested (Figure 9).

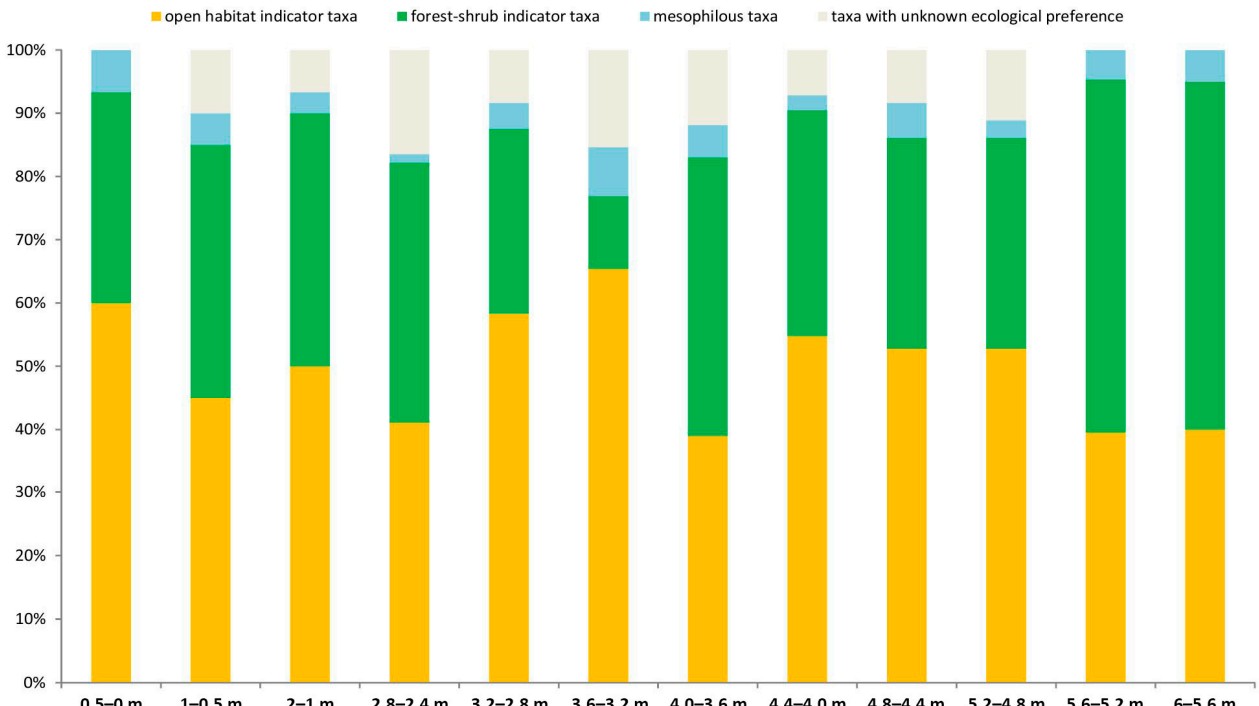

**Figure 8.** Percentage distribution of small mammal fauna by ecological preference in the stratigraphic sequence of the Süttő 21 site. The taxa with different habitat preferences are listed in Table 4.

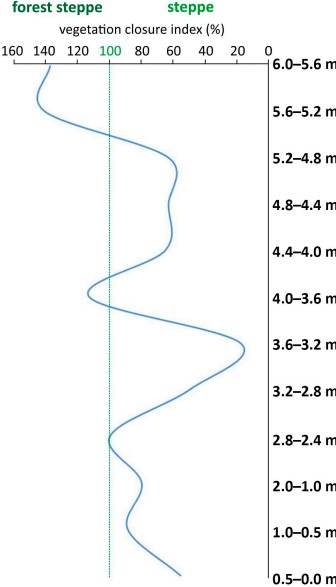

**Figure 9.** Changes of the vegetation closure index in the stratigraphic sequence of the Süttő 21 site. The vegetation closure index is 100 if $MNI_{forested} = MNI_{open}$, less than 100 if the vegetation was more open, and greater than 100 if the vegetation was more forested.

Three more closed and two more open periods of vegetation can be distinguished in the sequence. Figure 8 shows that the vegetation became more closed as it moves toward the top of the sequence. While at the bottom of the infilling (0.5–2.8 m), the proportion of those taxa preferring forest–shrub vegetation was 80–100% of those taxa preferring open habitat, in the next wave (3.6–4 m), this proportion rose to 113%, while at the top of the sequence (5.2–6 m), it reached 141%. In the other phases, we can imagine a more open steppe environment with shrubby vegetation limited to the area around the water body.

Given the total absence of *Crocidura* species typical of the warm steppe, this environment could have been a cold-climate steppe or forest-steppe throughout.

The taxonomic shifts discussed in the biostratigraphic section are also reflected in ecological features in the upper part of the sequence. The dominance of *Sorex runtonensis*, *Lagurodon arankae*, and *Prolagurus pannonicus*, which are specifically cold steppe species, is reduced, and the already present mesophilous shrews and voles (*Sorex minutus*, *Asoriculus gibberodon*, *Allophaiomys praehintoni*, *Lasiopodomys hintoni*, *Microtus* (*M.*) *nivaloides*) are joined by a new *Sorex* species, probably also in forested vegetation. The presence of a *Talpa* species confirms the increased proportion of forested or scrubby areas at the top of the sequence, as moles prefer the looser soils of forest or scrub.

As the proportion of different steppe species varies widely across the series, we attempted to provide a more accurate reconstruction of how open area characteristics have changed, based on recent analogies. To do this, we classified the taxa according to their habitat into four categories: arid or semi-arid grassland (*Lagurodon arankae*, *Prolagurus pannonicus*, *Sorex runtonensis*), fertile lowland steppe (*Allocricetus bursae*, *Cricetus cricetus* ssp.), shortgrass-steppe (*Spalax* sp., *Spermophilus* cf. *primigenius*), and forest-steppe (*Sicista* cf. *praeloriger*, *Ochotona* sp.) (Figure 10).

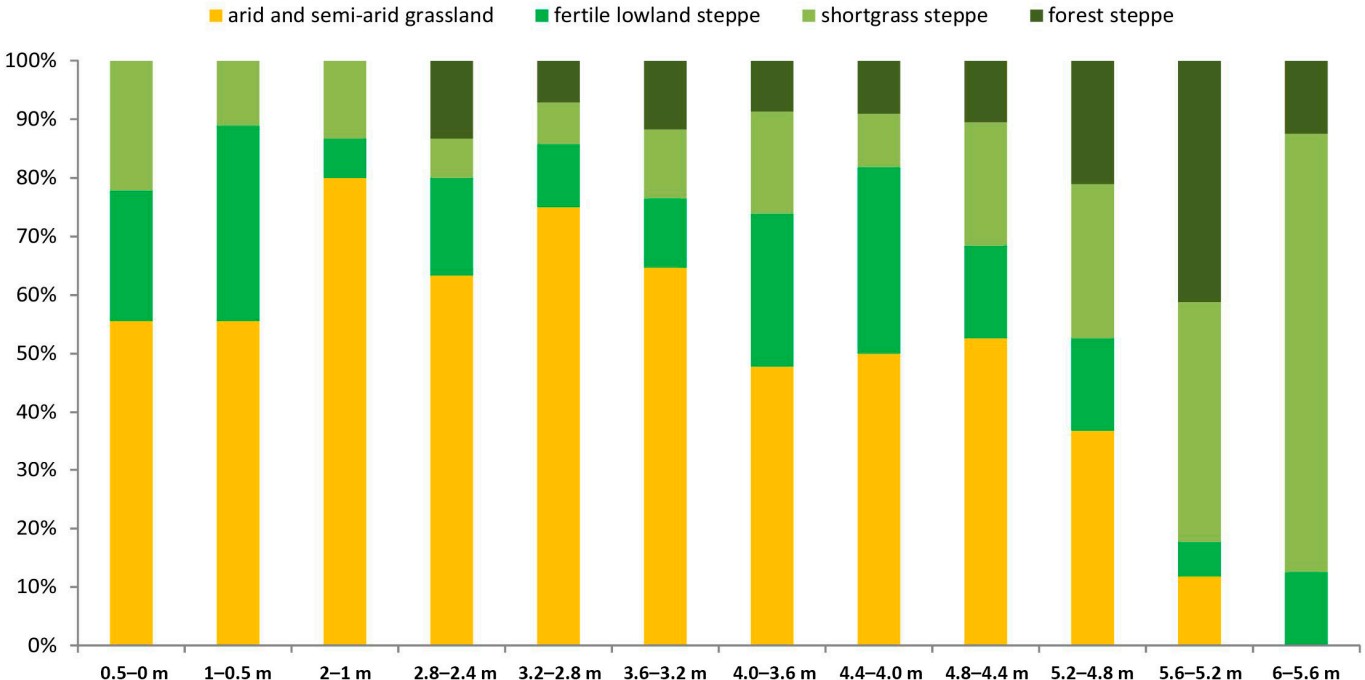

**Figure 10.** Percentage distribution of the so-called steppe taxa by habitat in the stratigraphic sequence of the Süttő 21 site. Arid or semi-arid grassland—*Lagurodon arankae*, *Prolagurus pannonicus*, *Sorex runtonensis*. Fertile lowland steppe—*Allocricetus bursae*, *Cricetus cricetus* ssp. Shortgrass-steppe—*Spalax* sp., *Spermophilus* cf. *primigenius*. Forest-steppe—*Sicista* cf. *praeloriger*, *Ochotona* sp.

Figure 10 shows that, during the deposition of the lower part of the sequence (0–4 m), the area around the site was mainly cold, dry steppe, and then from about 3.6–4 m upwards, a higher proportion of more forested steppe types with more precipitation appeared. This supports the results of the paleoecological analysis of the entire small mammal fauna and may explain the disappearance of some species of voles and shrews in the upper part of the series.

The large mammal remains indicate a mosaic environment with mainly open, grassy-steppe vegetation, which may have been interspersed with patches of forest. Compared to other localities, the strikingly few finds of small carnivores (mustelids) and the complete absence of small-sized large carnivores (canids, i.e., wolf and fox) in the large mammal

material is a very strange feature. The absence of the canids, combined with the relatively high number of remains from juvenile herbivorous large mammals (especially cervids), suggests that these phenomena are probably due to the presence of a large predator (i.e., the sabre-toothed *Homotherium*).

## 5. Comparison of the Süttő 21 Site with Sites of a Similar Age in Hungary

It has been known for a long time that significant ecological differences can be detected between the northern and southern parts of the Carpathian Basin, just as it is now, starting from the Pliocene [2]. Exploring these differences is difficult because the periods in which vertebrate faunas are known from both areas are rare. In addition to the relatively rich small and large vertebrate material, the value of the site is also enhanced by the fact that it provides well-documented information on a Pleistocene interval known from only a few sites in Hungary. The fauna of the Süttő 21 stratigraphic sequence in Gerecse Mountains (Northern Hungary) can be compared well with the material from another site of Süttő (Süttő 17) [3], the sites of Somssich Hill 2 and Villány 8 in the Villány Hills (Southern Hungary), Kövesvárad in the Bükk Mountains, and Újlaki-hegy in the Buda Hills (Northern Hungary), which are of a similar age, thus revealing taxonomic and paleoecological differences between different areas of the country [9]. The stratigraphic position of these sites used in the comparison is shown in Figure 5.

When comparing the sites in Northern and Southern Hungary, the first thing that stands out is the taxonomic differences. The first two of the species that play an important role in the vole fauna of the Villány Hills (*Allophaiomys pliocaenicus*, *Terricola arvalidens*, *Lasiopodomys hintoni*, *Microtus* (*M.*) *nivaloides*) are completely absent from the Northern Hungarian sites. Instead of *Allophaiomys pliocaenicus*, *Allohaiomys praehintoni* is present everywhere in the northern sites, along with *Lasiopodomys hintoni* and *Microtus* (*M.*) *nivaloides*.

The taxonomic differences are clearly due to ecological differences between the two areas. While in the Villány Hills, a generally warm, dry climate and open vegetation (probably the most-closed vegetation is karst shrubland) [6] can be reconstructed in the late Early Pleistocene and Early/Middle Pleistocene; the Northern Hungarian areas had a cooler, wetter climate and slightly more forested vegetation (sparse forest, forest-steppe) [3]. The different ranges of the two *Allophaiomys* species may be explained by the different climates and environments. *Allophaiomys pliocaenicus* may have been a warm, dry steppe species, while *Allophaiomys praehintoni* may have been a species of vole living in a cooler, wetter climate in forest-steppe vegetation.

## 6. Discussion and Conclusions

The rich vertebrate material of the Süttő 21 site provides a better understanding of the taxonomic and environmental changes around the Early/Middle Pleistocene boundary. A relatively dense sampling every 40 cm in the 6 m-high continuous sequence of the site has allowed both the discovery of taxonomic changes within the sequence and the paleoecological studies that have allowed the reconstruction of environmental changes during the deposition of the sediment.

Based on the zonal index species (*Mimomys pusillus*, *M. savini*) and some species of shrews and voles (*Beremendia fissidens*, *Asoriculus gibberodon*, Sorex *runtonensis*, *Lagurodon arankae*, *Allophaiomys praehintoni*), the lower part (0–4.4 m) belongs to the Early Pleistocene *Mimomys pusillus–Mimomys savini* Assemblage Zone, whereas the upper part (4.4–6 m) probably belongs to the *Mimomys savini* Biozone.

The taxonomic changes at the presumed Early/Middle Pleistocene boundary may have been caused by environmental changes that can be well reconstructed by paleoecological studies of vertebrate fauna. In general, the vegetation became more forested as it moved up the stratigraphic sequence, as indicated by changes in the proportions of steppe taxa and forest–shrub environment taxa, and by an increase in the proportion of grassland–shrubland and open forest species within the steppe species and the disappearance of dry steppe species. The same changes suggest that the climate became more humid, with

higher levels of precipitation towards the top of the stratigraphy, but the complete absence of *Crocidura* shrew species suggests that this change was not accompanied by warming. This could be interpreted as a change from the dry, cold steppe vegetation at the bottom of the sequence to forest-steppe vegetation developed in also cold climates at the top of the series.

A comparison of the Süttő 21 site and other Hungarian vertebrate sites of a similar age (between 1.1–0.7 Ma) revealed taxonomic and paleoecological differences between the northern and southern parts of the country. In contrast to the warm, dry climate and steppe vegetation typical of the Villány Hills, the northern part of the country (Gerecse, Buda Hills, Bükk) was characterized by a cool, wet climate and more closed (sparse forest) vegetation in the late Early Pleistocene and Early/Middle Pleistocene. For species occurring exclusively in one area, this allowed autecological conclusions to be drawn.

Of the vole species belonging to the *Microtus* evolutionary lineage, only *Lasiopodomys hintoni* and *Microtus* (*M.*) *nivaloides* were previously assumed to prefer cooler, wetter climates and semi-enclosed vegetation and *Terricola arvalidens* to have a clear preference for warm, dry climates and open vegetation [6], while the ecological requirements of the two *Allophaiomys* species were not clearly understood. However, the current work helps to clarify the ecological preferences of the two *Allophaiomys* species. The co-occurrence of *Allophaiomys pliocaenicus* with *Terricola arvalidens* at the Somssich Hill 2 site and its absence from the Süttő 21 site suggests that this species lived in warm, dry climates and open environments. In contrast, the present work suggests that *Allophaiomys praehintoni* preferred cool, moist climates and semi-enclosed (forest-steppe) environments. Their different ranges are, therefore, due to ecological differences.

**Author Contributions:** P.P. and M.G. conceptualised the idea and designed the methodology along with Z.S. All authors collected and analysed the data. The original draft was written by P.P., while the manuscript was reviewed and edited by Z.S., L.M., J.H. and M.G. All authors contributed to the drafts and gave final approval for publication. All authors have read and agreed to the published version of the manuscript.

**Funding:** This research received no external funding.

**Institutional Review Board Statement:** Not applicable.

**Data Availability Statement:** The data presented in this study are available in the article.

**Acknowledgments:** We thank Tibor Adamcsik, and his staff for their help during the fieldwork. We would like to thank Attila Virág, Sándor Krizsán, Eszter Hankó, Bence Szabó, Zsófia Tischner, Dalma Kerekes, Sándor Béres, and Zoltán Tóth for their help in the excavation works. We also owe many thanks to Dóra Borka and Ákos Juhász, as well as the students of Eötvös Loránd University, the University of Pécs, and the University of Szeged for their help with washing the sediment collected and sorting the remains. We would especially like to thank Márton Szabó for the photo plates. The latter were doing their internship at the Department of Palaeontology and Geology. This paper is ELKH-MTM-ELTE Paleo Contribution No. 373.

**Conflicts of Interest:** The authors declare no conflict of interest.

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
