# Peer review of "Stratigraphic and Paleoecological Significance of the Early/Middle Pleistocene Vertebrate Fauna of the Süttő 21 Site"

_diversity, doi:10.3390/d15060736_

Round 1

Reviewer 1 Report

Interesting paper on a new Pleistocene site from the Sütö area, especially when fossil record stems from a long-term fissure, which are filled often in very short time...

All my comments are highlighted directed in the attached file. One "problem" what I see, is absence of magnerostratigraphy ... Also, the last chapter could be called Discussion and Conclusion...

Reviewer 2 Report

A concise report on a new site with exciting faunal record ilustrating a transition period of early and late Biharian, one of the least known stage of the Quaternary faunal history. The fauna resembles that from Untermassfeld (which position is fixed to Jaramillo event  by magnetostratigraphic record). A brief discussion concerning a comparison between  the fauna under study and that from Untermassfeld (Maul 2001) would be greatly appreciated.  

The difference seems to be in absence of lagurins in the latter site (which may suggest a real  paleobiogeographic restriction of the lagurins range to Pannonian basin in the respective stage), and supposedly in a different interpretation of Microtus: Microtus thenii (including hintoni morphotypes and rare  "Allopahiomys praehintoni" morphotypes) in Untermassfeld, while a distinct separation of "Allophaiomys praehintoni" and "Terricola hintoni"  suggested in the present paper. Of course, appearance of Microtus nivaloides / nivalinus indicates clearly for Sütte 21 a more advanced stage of Microtus evolution than that demonstrated in Untermassfeld. At least a brief comment on such topics could supplement the present paper in a proper way. I personally would hesitate a bit of an implicite regarding hintoni as Terricola - relations to gregaloides-gregalis seems to more likely for it  (i.e. Lasiopodomys or Stenocranius if you wish, seems to be more appropriate). For that reason I personally would prefer not to give a strict generic assigment for early forms of  Microtus s.l. but refer them as Microtus spp. Yet, of course, it is entirely upon the authors which of the concepts they will prefer. Detailed analyses of these topics would largely exceed scope of the present paper. 

As a report of an exciting new site, its diversified fauna (including a rich amphibian and reptile record) and detailed paleoecological interpretation the present paper is a valuable contribution worth of publishing without essential changes.

In more loci the taxon names are not in itallics - please correct it.

Reviewer 3 Report

In general, this is a very informative and professionally written paper on a new faunal sequence of late Early Pleistocene age. In fact it is the period of the broadly discussed mid-Pleistocene climatic revolution. Would you add several sentences to put your data into a wider European and global climatic context?

The main issue of is the weak discussion of correlation of the new site with similar reference sites in Europe.

General taxonomy. I am aware that the taxonomy used reflects traditional/national concepts of the authors and in this aspect is not critical to the validity of the main results of the study. Anyway I would remark that the usage of Terricola (a well defined genus of grey voles, for a group of voles not identical to any fossil vole with T4-T4 fused) with Stenocranius variability may be a mere typology. Likevise Allophaiomys praenintoni may be a typological fragment of the continuous variability range of the gregaloid or arvaloid voles. Without an analysis of M3 it is difficult to understand which group or a combination of groups you have.

To characterize thу biochronological position of the SH 21 sequence it would be very informative to give a short account of Prolagurus pannonicus voles. Do you have predominant simple, rounded anteroconid caps present, or transylvanicus morphotypes also present?

The excellent quality figure of the vole identified as Microtus nivalinus deserves a short discussion of the lineage attribution of this form, comparison with the type material (it is dissimilar) and its importance for the biochronology.

I would give a table illustrating main arvicolid types (of the modal classes in case of variability ranges) of the sequence.

Mimomys savini - Mimomys pusillus zone vs Mimomys savini zone. I may refer and you should have known, that in Eastern Europe (Don basin and sw Urals) Mimomys pusillus clearly survives into the early Middle Pleistocene thus rendering the Mimomys savini - Mimomys pusillus zone diachronous.

Please discuss a possibility of the attribution of the smaller Cricetus cricetus to C. nanus Schaub.

Paleoenvironmental part

Erinaceidae gen. et sp. can easily represent a forest dwelling hedgehog.

Pliomys episcopalis are typically mesophilic taxon easily attributable to the forest ecological group.

Fig. 8: I would revers thу vertical axis to make older/deeper level of the section lower to fit the stratigraphic logical view.

Please check throughout the text the italization of the Latin names.

page 8. "Mimomys savini" .. "the top of the sequence is characterized by rootless, large forms" do you mean completely rootless forms, i.e. early Arvicola? Or just juvenile specimens with early stages of root formation?

page 13. "water-related shrews and voles (Sorex minutus, Asoriculus

gibberodon, Allophaiomys praehintoni, Terricola hintoni, Microtus (M.) nivaloides"

I would change "water-related" to "mesophilous"

page 15. Figure 10. Please indicate in the captions that the vertical line is the presumed chronological range of the studied site SüttÅ‘ 21.

"forested vegetation (grove, forest-steppe)" please define "grove" as a type of the forest, "sparse forests"?

page 16: "more precipitated, but also cold forest-steppe vegetation" -->

"a vegetation of cold forest-steppe type indicating higher precipitation"

"climate has become increasingly precipitated" by this wording you say that climate deteriorates, but instead you probably mean that the climate became more humid with higher level of precipitation. If valid please correct.

"Based on the current work, Allophaiomys pliocaenicus inhabited warm, dry climates and open environments"

but you do not report Allophaiomys pliocaenicus in you material and the analysis.

The English is generally clear and neat. A couple of issues is indicated.
